Microbiology **Spectrum**

# Protein-mediated stabilization of amphotericin B increases its efficacy against diverse fungal pathogens

Kenya E. Fernandes,[1,2] Caitlin L. Johnston,[3] Brayden C. Williams,[3] Dee A. Carter,[1,2] Margaret Sunde[2,3]

**ABSTRACT** Amphotericin B (AMB), a potent and broad-spectrum antifungal agent, faces solubility and toxicity challenges in clinical use. In this study, we explored the ability of DewY and $EAS_{\Delta 15}$, class I fungal hydrophobin proteins with unique amphipathic properties and self-assembly capabilities, to stabilize AMB in solution. UV-visible spectroscopy confirmed the ability of hydrophobin proteins to stabilize the monomeric state of AMB in aqueous solution for up to 48 h. Further assays revealed that this effect was not exclusive to hydrophobins, however, as non-hydrophobin proteins provided similar stabilizing effects. AMB-protein combinations exhibited enhanced efficacy against diverse clinically relevant fungal pathogens, with 4- to 32-fold reductions in the effective *in vitro* dosage compared to AMB alone. Microscopic analyses found fungal cells treated with AMB alone and in combination with proteins had identical morphological changes, suggesting that protein interactions do not alter the mode of action of AMB. Instead, our results indicate that the monomeric state of AMB is stabilized in aqueous solution by non-specific interactions with hydrophobic areas on proteins. We suggest that this protein-mediated enhancement of solubility could reduce the required dose of AMB, providing a basis for optimizing AMB-based antifungal therapies.

**IMPORTANCE** Fungal infections are a growing global health concern, yet effective antifungal treatments remain limited by toxicity and poor solubility. AMB, a potent broad-spectrum antifungal, is highly effective but suffers from severe side effects and formulation challenges. Our study demonstrates that proteins, including fungal hydrophobins, can stabilize AMB in its monomeric form, significantly enhancing its solubility and efficacy against a range of fungal pathogens. These findings suggest that protein-mediated stabilization could enhance the effectiveness of AMB by reducing the required dosage and potentially lowering its toxic side effects. This approach offers a promising strategy for optimizing AMB therapies and improving treatment options, especially in resource-limited settings where fungal infections impose a significant health burden.

**KEYWORDS** amphotericin B, hydrophobins, antifungals, combination therapy

Address correspondence to Kenya E. Fernandes, kenya.fernandes@sydney.edu.au.

Kenya E. Fernandes and Caitlin L. Johnston contributed equally to this article. The order of names was determined alphabetically.

The authors declare no conflict of interest.

See the funding table on p. 14.

Amphotericin B (AMB) is a potent polyene antifungal drug used to treat life-threatening fungal infections. Its broad-spectrum efficacy arises from its ability to bind ergosterol in fungal membranes, disrupting membrane integrity (1). However, the clinical use of unmodified AMB is hampered by severe nephrotoxicity and poor aqueous solubility (2). Formulation with sodium deoxycholate improves solubility but is still associated with dose-limiting toxicities. Lipid- and liposomal-based formulations have been developed to mitigate toxicity and improve delivery. However, while these formulations have improved therapeutic outcomes, their very high production cost restricts their accessibility (3, 4). Recently, research into AMB analogs has identified new

renal-sparing derivatives, although challenges related to formulation and production costs are likely to persist (5).

The growing global burden of fungal infections, exacerbated by the increasing immunocompromised population and the potential emergence of novel pathogens linked to global change, underscores the urgent need for accessible and effective antifungal therapies (6). This need is particularly acute in resource-limited regions, where fungal infections impose significant morbidity and mortality. Addressing this requires new and affordable broad-spectrum antifungal solutions. One promising approach is synergistic therapy, which combines two or more agents to enhance efficacy while lowering drug doses, thereby reducing toxicity and off-target effects. Recent studies have shown the potential of AMB combined with various agents, including other antifungal drugs (7), repurposed medications (8), proteins (9), and peptides (10). These findings provide promising leads for therapeutic development and can deepen our understanding of the molecular mechanisms underlying the biological activity of AMB (4).

A key challenge in optimizing AMB therapy lies in its propensity to aggregate and present as both monomeric and multimeric forms, each with distinct biological properties (11). Studies have shown that AMB aggregates are toxic to both erythrocytes and fungal cells, while monomers are selectively toxic to fungal cells (12, 13). Modifying AMB to increase its critical aggregation concentration has shown potential for reducing hemolytic activity and increasing selectivity for fungal cells (14). Therefore, exploring combinatory agents that could reduce aggregation and stabilize the monomeric form of AMB in solution may offer a promising strategy to improve its therapeutic profile.

Fungal hydrophobins are a family of small, amphipathic proteins produced exclusively by fungi. Hydrophobins self-assemble into monolayers that modify interfaces, contributing to fungal biology by making surfaces hydrophobic and non-immunostimulatory (15–17). These proteins exhibit a large exposed hydrophobic surface area, which underpins their surface activity and ability to interact with hydrophobic molecules (18–20). Class I hydrophobins form amyloid-structured fibrils known as rodlets, which pack into amphipathic monolayers (17, 18). Notably, hydrophobins are non-immunostimulatory in their isolated, purified form, showing no activation of dendritic cells, alveolar macrophages, or helper T-cell immune responses (16) and are widely consumed in edible mushrooms without adverse effects (21). These attributes have led to significant interest in their potential medical and biotechnological applications. Hydrophobins have been shown to stabilize hydrophobic compounds in solution and to improve the stability of drug-loaded liposomes (15, 22, 23). They have also been shown to be non-cytotoxic across different cell types and upon intravenous delivery (24, 25). They could therefore offer a novel approach to overcoming the challenges associated with hydrophobic drugs like AMB.

We have previously reported that the class I hydrophobin DewY (from *Aspergillus nidulans*) stabilizes AMB in aqueous solution as a rodlet-coated nanoparticle form (26). Here, we sought to evaluate how DewY and another class I hydrophobin, $EAS_{\Delta 15}$ (from *Neurospora crassa* [19]), might affect the antifungal activity of AMB. We found that these hydrophobins effectively stabilized AMB in solution and increased its potency. However, our investigation of the physicochemical basis for this interaction revealed that other proteins could stabilize AMB, suggesting that the effect may stem from transient, non-specific interactions with hydrophobic areas on protein surfaces. Our results suggest that protein-mediated stabilization of AMB could enhance its therapeutic profile by improving solubility and reducing the dose required for effective antifungal activity.

## MATERIALS AND METHODS

### Fungal strains and culture conditions

Three yeast species (*Saccharomyces cerevisiae* S288c, *Cryptococcus neoformans* H99, and *Candida albicans* SC5314) and two mold species (*Aspergillus fumigatus* ATCC204305 and *Mucor circinelloides* 80–14-240-5007) were used for antifungal testing. Strains were maintained as glycerol stocks at −80°C, grown on potato dextrose agar (Oxoid), and incubated at 30°C for 24–48 h before use.

### Protein production and purification

Hydrophobin proteins DewY and EAS$_{\Delta15}$ were produced recombinantly as His$_6$-ubiquitin-tagged fusion proteins, with EAS$_{\Delta15}$ purified as previously described (26). DewY was purified under native conditions via Ni-NTA affinity chromatography, followed by cleavage from the fusion partner, and finally purification by reverse phase high-performance liquid chromatography. Purified DewY and EAS$_{\Delta15}$ were lyophilized and stored at −20°C until use. The cleaved His$_6$-ubiquitin (Ub) produced as a by-product of DewY purification was stored under the same conditions and used subsequently for AMB + Ub formulations. Bovine α-lactalbumin (α-lac, L6385), bovine serum albumin (BSA), and thioflavin T (ThT) were purchased from Sigma-Aldrich (St. Louise, MO, USA).

### Formulation of drug and protein samples

AMB, fluconazole (FLC, Sapphire Bioscience) and nystatin (NYS, Sigma-Aldrich) were solubilized in DMSO, and the deoxycholate formulation of AMB (AMB-D, Sigma-Aldrich) and caspofungin (CAS, Sigma-Aldrich) were solubilized in MilliQ water at 100 µg/mL. Lyophilized DewY, EAS$_{\Delta15}$, Ub, BSA, and α-lac were dissolved in MilliQ water at a concentration of 50 or 100 µg/mL. For drug-protein samples, the drug master stock was added to solubilized protein solution to achieve a final drug concentration of 1 µg/mL (1% dimethyl sulfoxide [vol/vol]) in glass vials. All drug-protein samples were mixed gently, except for the AMB shaken/rodlet samples, which were vigorously mixed for 1 h to promote the assembly of DewY rodlets at air-water interfaces.

### AMB stability monitoring by UV-visible spectroscopy

AMB stability in the presence or absence of proteins was assessed using UV-visible (UV-vis) spectroscopy. Following preparation, AMB formulations were transferred to a cuvette, and absorbance spectra were immediately measured (300–450 nm) using a Beckman DU 800 spectrophotometer (Beckman Coulter). Samples requiring 1 h incubation at room temperature were placed back into vials, and the spectra were collected again after the appropriate time.

### ThT fluorescence kinetic assays

Lyophilized DewY was dissolved in MilliQ water at 50 µg/mL and added to a 96-well plate with 40 µM ThT in the presence or absence of AMB, CAS, or FLC at 1 µg/mL. Rodlet assembly was initiated by vigorously shaking the plate at 700 rpm in a POLARstar Omega plate reader (BMG Labtech). Fluorescence intensity was measured at room temperature with constant shaking over 100 min using excitation/emission filters of 440/480 nm, respectively.

### Transmission electron microscopy

Samples containing DewY in the presence and absence of AMB were retrieved from the ThT assay, and 10 µL of each was floated on standard Formvar on carbon film-coated transmission electron microscopy grids (ProScitech) for 4 min. Excess liquid was removed by touching the grid to filter paper. Grids were washed with water and stained by floating on 2% uranyl acetate solution for 4 min, then air-dried. Grids were examined and imaged using an FEI Tecnai T12 microscope operating at 120 kV.

## Antifungal susceptibility testing by broth microdilution

Broth microdilution was performed in 96-well plates following Clinical and Laboratory Standards Institute guidelines for yeasts (M27-A3) and filamentous fungi (M38-A2). Fungal inocula were prepared from colonies to a final concentration of $0.5 \times 10^3$ to $2.5 \times 10^3$ CFU/mL for yeasts and $0.4 \times 10^4$ to $5 \times 10^4$ CFU/mL for molds. All tests used RPMI-1640 (Sigma-Aldrich) supplemented with 0.165 M 3-(N-morpholino)propoanesulfonic acid (MOPS) and 2% D-glucose, except for *C. neoformans*, where YNB (Sigma-Aldrich) supplemented with 0.165 M MOPS and 0.5% D-glucose was used. Assays in serum used media supplemented with either 10% human or fetal bovine serum (FBS, Sigma-Aldrich). Higher serum concentrations (50% and 90%) were tested but found to inhibit fungal growth too extensively for reliable minimum inhibitory concentration (MIC) determination. Plates were incubated at 35°C for 24 h (*S. cerevisiae* and *C. albicans*) or 48 h (*C. neoformans*, *A. fumigatus*, and *M. circinelloides*). The MIC was determined as the lowest drug concentration that inhibited 100% growth and was read visually or by absorbance at 600 nm (BioTek Elx800) for assays with human serum. Assays were performed at least twice, and reported values are the mode. For time-kill curves, CFU per milliliter was measured at 0, 3, 6, and 12 h by backplating on agar plates.

## Drug interaction testing and modeling

Diagonal-sampling checkerboard assays were used to determine pairwise interactions between AMB and test proteins (27). Full checkerboard assays were subsequently performed on AMB in combination with DewY, EAS$_{\Delta 15}$, and Ub against *S. cerevisiae*. Serial twofold dilutions of AMB and each interacting protein were prepared in 96-well plates in the horizontal and vertical directions, respectively. Inoculum preparation, media, and incubation conditions followed the conditions for antifungal susceptibility testing outlined above. Results were assessed visually and by absorbance at 600 nm (BioTek Elx800). Fractional inhibitory concentration index (FICI) was calculated based on the Loewe additivity model as fractional inhibitory concentration (FIC) = (MIC$_X$ / MIC$_Y$), where MIC$_X$ is the MIC of the drug alone and MIC$_Y$ is the MIC of the drug in combination. FICI is the sum of the FIC for each drug. Interactions were classified as synergistic (≤0.5), additive or indifferent (>0.5–4.0), or antagonistic (>4.0).

## Assessment of microscopic morphology following treatment

*C. albicans*, *C. neoformans*, *A. fumigatus*, and *M. circinelloides* cultures were untreated or were treated with 1/2 or 1/4 MIC or FIC combinations of AMB, DewY, Ub, AMB + DewY or AMB + Ub. Inoculum preparation, media, and incubation conditions followed the conditions for antifungal susceptibility testing outlined above. At 24 h (*C. albicans*) or 48 h (*C. neoformans*, *A. fumigatus*, and *M. circinelloides*), test wells were examined and photographed using an IS10000 Inverted Microscope (Luminoptic) at 40× using ISCapture Imaging software (Tucsen Photonics).

## Calculation and visualization of exposed surface patches on the proteins

Surface hydrophobic maps of the proteins DewY (PDB ID: 2LSH), EAS$_{\Delta 15}$ (PDB ID: 2FMC), Ub (PDB ID: 1UBQ), α-lac (PDB ID: 1F6S), and BSA (PDB ID: 3V03) were generated, and the percentage of exposed hydrophobic patches on the surface of the proteins was calculated using Swiss PDB Viewer (https://spdbv.unil.ch/) (25, 28). Image showing the surface hydrophobicity for DewY was generated using ChimeraX with the surface displayed according to molecular hydrophobicity potential.

## RESULTS

### Hydrophobins and ubiquitin increase the solubility of AMB in solution

The distinctive absorbance peaks of AMB were utilized to monitor its solubility and self-association: peaks at 409, 385, and 365 nm reflect the presence of monomeric AMB,

while a peak at 348 nm is characteristic of AMB aggregates (29, 30). We employed UV-vis spectroscopy and monitored the ratio between the monomer peak at 409 nm and the aggregate peak at 348 nm ($A_{348/409}$) to determine AMB solubility and aggregation in water across increasing drug concentrations (1–32 µg/mL). At all concentrations tested, absorbance spectra consistently displayed the monomer and aggregate peaks (Fig. 1a). As expected, the amplitude of all peaks increased with higher AMB concentrations, with a relatively greater increase in the aggregate peak at higher concentrations. An aggregation ratio ($A_{348/409}$) of <1 is characteristic of the monomeric form of AMB (31). The aggregate ratios increased from 0.76 at 1 µg/mL to 4.17 at 32 µg/mL, indicating a transition from monomers to aggregates at higher concentrations.

We assessed the ability of the hydrophobin proteins DewY and EAS$_{\Delta 15}$ to stabilize AMB in the monomeric state. His$_6$-Ub was included as a control protein. UV-vis measurements were collected for 1 h from 1 µg/mL AMB solutions prepared with or without protein (Fig. 1b and c). At this concentration and in the absence of any protein, AMB was monomeric but poorly soluble, as indicated by comparatively low peak absorbance values, which decreased further over time due to the formation of insoluble aggregates. In the presence of both hydrophobin proteins, AMB remained soluble and stabilized in its monomeric form. DewY was more effective in stabilizing the monomeric form in solution, as indicated by a smaller decline in absorbance over time. Unexpectedly, the non-hydrophobin protein Ub also exhibited a similar monomer stabilizing and solubilizing effect, indicating that stabilization of the monomeric form of aggregation-prone AMB in aqueous solutions is not unique to hydrophobin proteins.

## Hydrophobins and ubiquitin enhance the efficacy of AMB

To determine whether the increased stabilization of the monomeric form of AMB in the presence of a hydrophobin protein or ubiquitin translated to a reduction in the inhibitory dose required for fungal cells, AMB-protein pairs were tested against *Saccharomyces cerevisiae* using a diagonal-sampling checkerboard assay with a starting concentration of 1 µg/mL AMB and 100 µg/mL protein. The resulting FICs are shown in Fig. 2a. The MIC of AMB alone was 1 µg/mL, and as expected, neither the hydrophobin proteins DewY

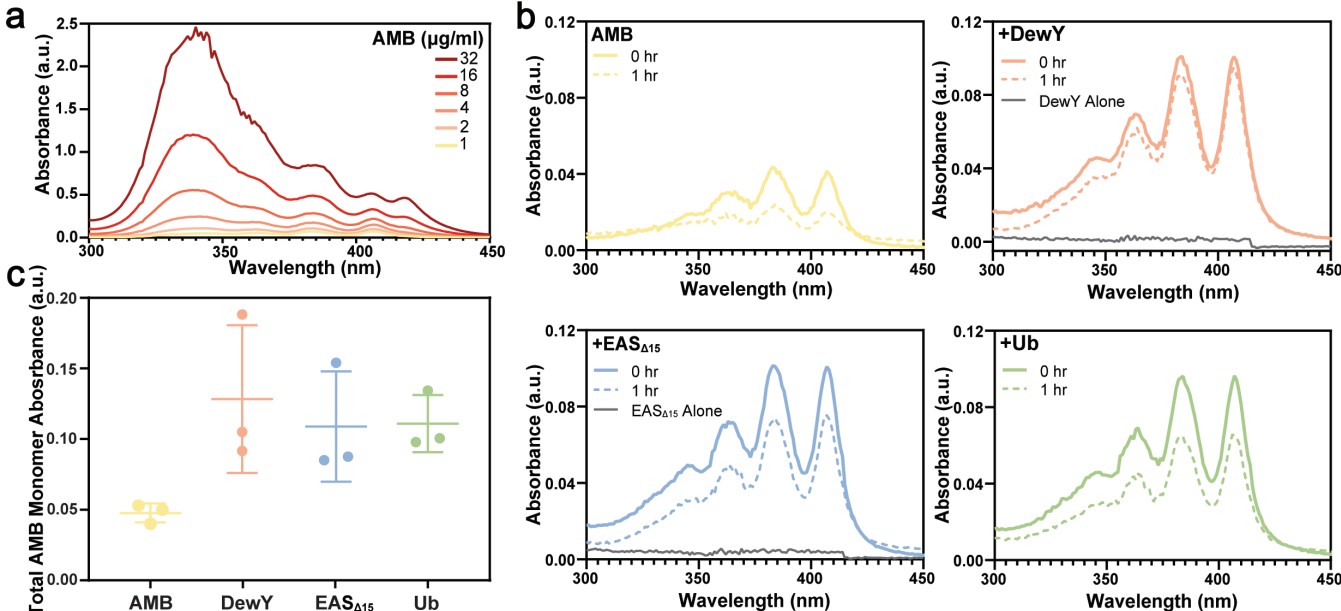

**FIG 1** Stabilization of AMB in the presence of the proteins DewY, EAS$_{\Delta 15}$, and Ub. (a) Absorbance spectrum of AMB in water at increasing concentrations (1–32 µg/mL) shows three monomer peaks (365, 385, and 409 nm) and an aggregate peak (348 nm). (b) Representative absorbance spectrum of AMB (1 µg/mL) in the presence or absence of DewY, EAS$_{\Delta 15}$, or Ub before and after 1 h incubation at room temperature. (c) Initial total absorbance of AMB monomers ($A_{365} + A_{385} + A_{409}$) in the absence (AMB) and presence of DewY, EAS$_{\Delta 15}$, and Ub shown as mean ± standard deviation ($n = 3$).

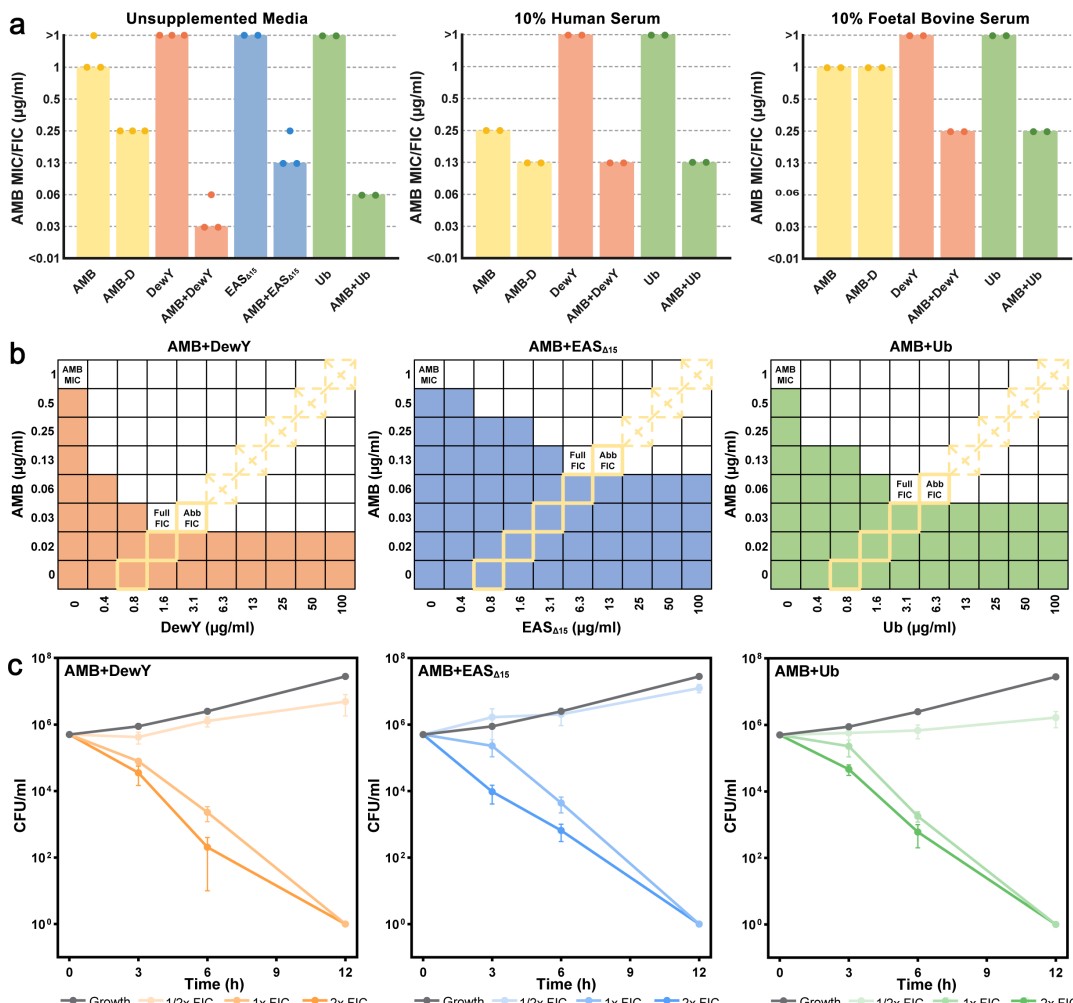

**FIG 2** Synergistic interaction of AMB with proteins in *Saccharomyces cerevisiae*. (a) The concentration of AMB required to inhibit growth when used alone or in the presence of DewY, EAS$_{\Delta15}$, or Ub determined using a diagonal-sampling checkerboard method (24). Data are shown for unsupplemented media (left) or media supplemented with 10% human (middle) or fetal bovine (right) serum. Minimum inhibitory concentration (MIC, the concentration of drug required to inhibit growth) is shown for AMB or AMB-D alone, while fractional inhibitory concentration (FIC, the concentration of drug required to inhibit growth in combination with another agent) is shown for AMB + protein pairs. Bars represent the mode of at least two biological replicates. (b) Full checkerboard assays performed to assess the interaction between AMB and DewY, EAS$_{\Delta15}$, or Ub. Colored cells denote growth, and white cells denote 100% inhibition. MIC values for each agent and FIC values for the combination of agents (full FIC) are indicated. DewY, EAS$_{\Delta15}$, and Ub had no achievable MIC. Concentrations previously tested using the abbreviated checkerboard method are denoted with yellow borders, with areas of growth denoted as dashed crosses, and the abbreviated FIC values (abb FIC) are indicated. (c) Time-kill assays of AMB in combination with DewY, EAS$_{\Delta15}$, and Ub over 12 h using the 1/2×, 1×, and 2× FIC determined via full checkerboard assays ($n = 3$). Error bars show the SEM.

and EAS$_{\Delta15}$ nor the control protein Ub alone affected fungal growth (MIC >50 µg/mL). AMB-D, which has a higher solubility in aqueous solution, had an MIC of 0.25 µg/mL, fourfold lower than AMB. All three AMB-protein pairs were found to be remarkably effective in combination, with AMB + DewY achieving a 32-fold reduction in the required concentration of AMB (FIC = 0.03125), AMB + EAS$_{\Delta15}$ achieving an 8-fold reduction (FIC = 0.125) and AMB + Ub achieving a 16-fold reduction (FIC = 0.0625).

To test the efficacy of these combinations in serum, which is a complex mixture containing numerous proteins, experiments were repeated with supplementation of either 10% human serum or FBS. In human serum, the MIC of AMB was reduced fourfold to 0.25 µg/mL, and AMB-D, AMB + DewY, and AMB + Ub achieved another twofold reduction. However, human serum alone reduced fungal growth by approximately 60% relative to controls, suggesting that these results may be confounded by other variables

that may alter the performance of AMB-protein pairs. Unlike 10% human serum, 10% FBS did not inhibit fungal growth, and the MICs of AMB and AMB-D were both 1 µg/mL. Under these conditions, AMB + DewY and AMB + Ub achieved a fourfold reduction in AMB (FIC = 0.25), indicating continued although somewhat reduced efficacy in the presence of other FBS components. Attempts to increase serum concentration to 50% and 90% resulted in near-complete fungal growth inhibition, making accurate MIC determination unfeasible.

Full checkerboard assays were next performed to explore the interaction profile of AMB-protein pairs further (Fig. 2b). These assays revealed that the same fold reduction in AMB concentration could be achieved with half the amount of each starting protein ("full FIC") compared to the tests using the abbreviated (abb) checkerboard assays ("abb FIC"). Time-kill assays performed using the updated FIC values (Fig. 2c) showed that AMB-protein pairs at 1× and 2× FIC resulted in 100% cell death within 12 h. At 1/2× FIC, fungal proliferation was delayed relative to the growth control.

## Protein-mediated potentiation is unique to AMB and is not observed with other antifungal drugs

To determine if these three proteins interact with other antifungals, DewY, $EAS_{\Delta15}$, and Ub were tested with the polyene NYS, the azole FLC, and the echinocandin CAS, in addition to AMB-D for comparison. The same diagonal-sampling checkerboard method was used with a starting concentration of 1 µg/mL AMB and 50 µg/mL protein. The FICIs for each combination, individual FICs, and AMB fold decrease are listed in Table 1. Interestingly, AMB-D was still potentiated by the three proteins but to a lesser degree (FICI = 0.125–0.5) compared to AMB (FICI = 0.0625–0.25). For NYS + DewY, the FICI was 1, indicating a twofold reduction in the NYS concentration required to achieve inhibition. Although a much smaller decrease than seen with AMB, this indicates that another drug in the same polyene drug class displays some interaction with this protein. All other NYS combinations and all FLC and CAS combinations achieved a no-fold reduction (FICI = 2), indicating no interaction.

## Monomer and rodlet forms of DewY interact with AMB to produce similar results

The DewY hydrophobin protein, which showed the greatest ability to stabilize monomeric AMB and achieved the largest fold reduction in AMB against fungal cells, was chosen for further detailed examination of the interaction with AMB. Class I hydrophobins such as DewY and $EAS_{\Delta15}$ are known to self-assemble and form amyloid fibril structures known as rodlets, and these further assemble into amphipathic monolayers that facilitate interactions between fungal structures (e.g., spores) and environmental or host surfaces (15, 18, 32). The rodlet assembly of hydrophobins can be monitored using the fluorescent dye ThT, which fluoresces upon binding to the cross-β structure characteristic of amyloid fibrils (18, 33). To assess the interaction of AMB with DewY during rodlet assembly, ThT fluorescence was measured in the presence and absence of AMB (Fig. 3a). When solutions of DewY were agitated, a marked increase in ThT fluorescence was observed, consistent with rodlet assembly. The addition of AMB reduced both the rate of increase and the final intensity of ThT fluorescence. The presence of CAS had a smaller effect, and FLC did not alter the ThT fluorescence profile. Transmission electron microscopy demonstrated that DewY rodlet formation occurred in both the presence and absence of AMB (Fig. 3b), suggesting that AMB prevents ThT from binding to the rodlet surface but does not inhibit DewY rodlet assembly. Similar effects have been observed with other low-molecular-weight molecules tested as inhibitors of amyloid assembly, where ThT binding is disrupted by competing compounds (34).

The effect of the different monomeric and rodlet forms of DewY on the stability of AMB was further compared by UV-vis spectroscopy and antifungal susceptibility testing. The absorbance spectra of AMB in the presence of DewY in monomeric and rodlet forms showed comparable high AMB absorbance peaks, indicating a similar stabilization

TABLE 1  Effect of commonly used antifungal drugs in combination with hydrophobins DewY and EAS$_{\Delta15}$ and human ubiquitin against *Saccharomyces cerevisiae*

| Protein | Drug FIC (µg/mL) | AMB fold decrease | Protein FIC (µg/mL) | FICI | Synergy |
|---|---|---|---|---|---|
| Amphotericin B | | | | | |
| AMB alone | 1 | – | – | – | – |
| +DewY | 0.03125 | 32 | 1.5625 | 0.0625 | Synergistic |
| +EAS$_{\Delta15}$ | 0.125 | 8 | 6.25 | 0.25 | Synergistic |
| +Ubiquitin | 0.0625 | 16 | 3.125 | 0.125 | Synergistic |
| Amphotericin B (deoxycholate)$^d$ | | | | | |
| AMB-D alone | 0.25 | – | – | – | – |
| +DewY | 0.015625 | 16 | 1.5625 | 0.125 | Synergistic |
| +EAS$_{\Delta15}$ | 0.0625 | 4 | 6.25 | 0.5 | Synergistic |
| +Ubiquitin | 0.03125 | 8 | 3.125 | 0.25 | Synergistic |
| Nystatin | | | | | |
| NYS alone | 8 | – | – | – | – |
| +DewY | 4 | 2 | 6.25 | 1 | Indifferent |
| +EAS$_{\Delta15}$ | 8 | 1 | 12.5 | 2 | Indifferent |
| +Ubiquitin | 8 | 1 | 12.5 | 2 | Indifferent |
| Fluconazole | | | | | |
| FLC alone | 8 | – | – | – | – |
| +DewY | 8 | 1 | 25 | 2 | Indifferent |
| +EAS$_{\Delta15}$ | 8 | 1 | 25 | 2 | Indifferent |
| +Ubiquitin | 8 | 1 | 25 | 2 | Indifferent |
| Caspofungin | | | | | |
| CAS alone | 16 | – | – | – | – |
| +DewY | 16 | 1 | 50 | 2 | Indifferent |
| +EAS$_{\Delta15}$ | 16 | 1 | 50 | 2 | Indifferent |
| +Ubiquitin | 16 | 1 | 50 | 2 | Indifferent |

$^a$FIC is shown for combinations; MIC is shown for antifungal drugs alone.
$^b$As proteins had no achievable MIC, the highest concentration tested (50 µg/mL) was used for the calculation of FICI.
$^c$≤0.5 = synergistic; >0.5–4.0 = indifferent; >4 = antagonistic.
$^d$Dashes where AMB was tested alone indicate no collection of fold-change data.

capacity (Fig. 3c). Antifungal susceptibility testing against *S. cerevisiae* showed that DewY in rodlet form (FIC = 0.625) was almost as effective as DewY in monomer form (FIC = 0.03125), achieving a 16-fold reduction in AMB (Fig. 3d). These results suggest that AMB binds similarly to DewY monomer and rodlets without impacting the self-assembly of the protein, and both stabilize the drug in its monomeric form.

## Stabilization of AMB can be achieved with a diverse range of proteins and persists for up to 48 h

We next tested additional proteins for the ability to stabilize AMB. BSA and α-lac were evaluated alongside DewY, EAS$_{\Delta15}$, and Ub. AMB monomer was monitored over 48 h by measuring the combined absorbance of monomer peaks ($A_{409} + A_{385} + A_{365}$) in the UV-vis spectra. All protein-AMB combinations initially showed higher monomer absorbance than AMB alone, but only DewY, EAS$_{\Delta15}$, and Ub showed sustained higher absorbance than AMB alone throughout the incubation period in all samples (Fig. 4a and b). The inclusion of DewY resulted in a total monomer absorbance of 0.156 at 0 h (compared to 0.095 for AMB alone) and 0.092 at 48 h (compared to 0.023 for AMB), demonstrating a significant and prolonged stabilization effect. The physicochemical basis for the stabilization of the AMB monomer by the proteins was explored by calculating and visualizing the total area of exposed hydrophobicity on their surfaces. Exposed hydrophobic patches make up 10% of the surfaces of BSA and Ub, 11% for α-lac, 16% for DewY, and 21% for EAS$_{\Delta15}$ (Fig. 4c). No clear relationship was observed between exposed hydrophobicity and the ability of each protein to stabilize the AMB monomer; however, DewY

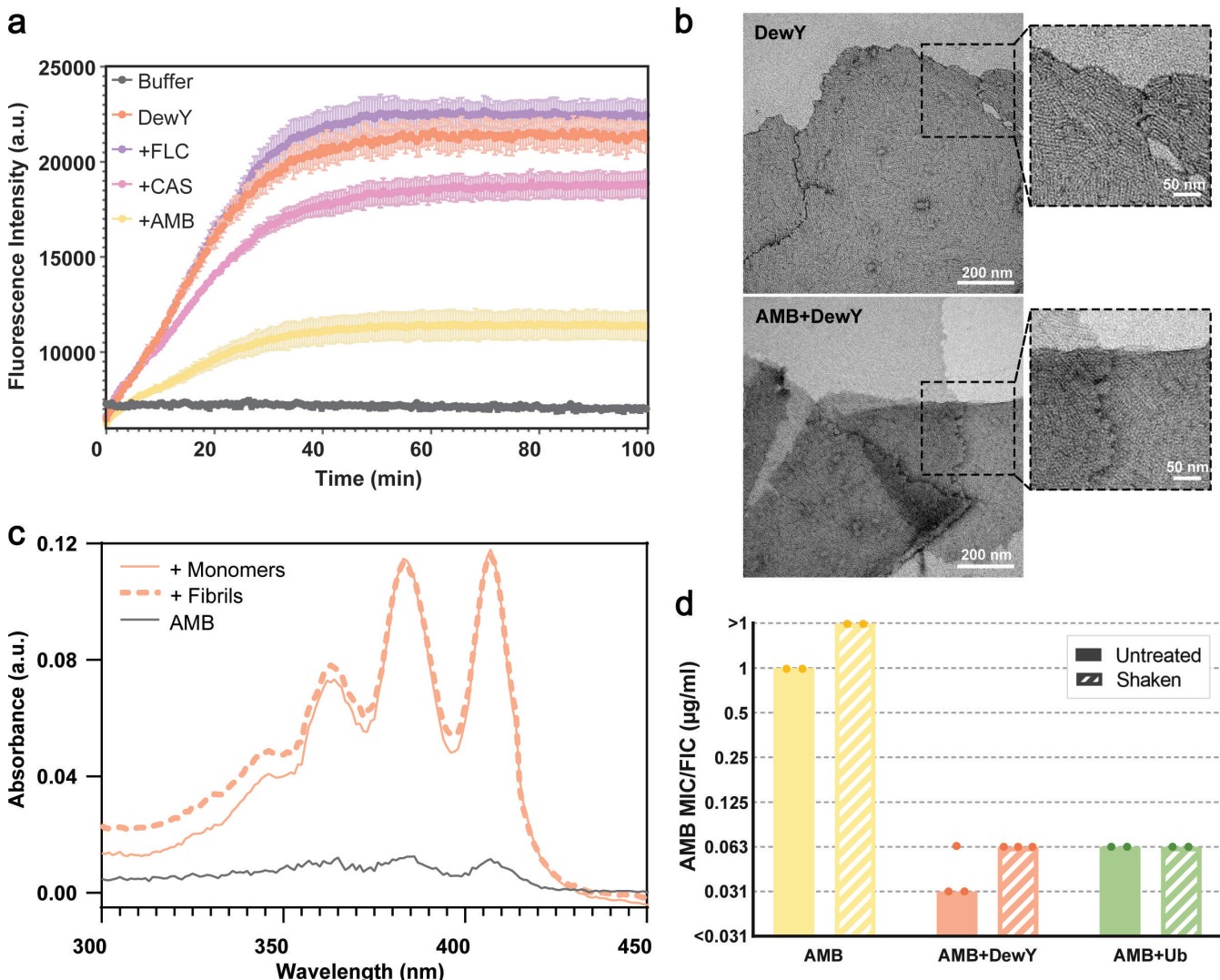

**FIG 3** Interaction of AMB with DewY rodlets. (a) Rodlet assembly of DewY (50 µg/mL) with shaking was monitored via ThT fluorescence in the presence or absence of FLC, CAS, or AMB. Three replicates of each sample are presented as mean ± standard deviation. (b) TEM images of ThT assay samples containing DewY in the presence and absence of AMB. Higher magnification images reveal the packing of rodlets to form monolayers. (c) Absorbance spectrum of AMB (1 µg/mL) in the presence of DewY monomer or rodlets. AMB binds to hydrophobin rodlets. (d) The concentration of AMB required to inhibit the growth of *Saccharomyces cerevisiae* alone or in the presence of DewY and Ub (100 µg/mL). Solid bars indicate untreated samples, and striped bars indicate shaken samples; all bars represent the mode of at least two biological replicates.

does exhibit a clear contiguous hydrophobic patch which may be the site of interaction with AMB (Fig. 4d).

Antifungal efficacy was assessed against four clinically relevant fungal pathogens: *Cryptococcus neoformans*, *Candida albicans*, *Aspergillus fumigatus*, and *Mucor circinelloides*. FICIs, individual FICs, and AMB fold-reduction values were determined using a diagonal-sampling checkerboard method with a starting concentration of 1 µg/mL AMB and 50 µg/mL protein (Table 2). All AMB-protein pairs showed improved efficacy compared to AMB alone against all species tested (FICI = 0.0625–0.5). Among the tested combinations, AMB + DewY was consistently the most effective (FICI = 0.0625–0.125), achieving a 32-fold reduction in AMB against *C. neoformans*, *C. albicans*, and *M. circinelloides* and 16-fold reduction against *A. fumigatus*. Conversely, AMB + EAS$_{\Delta 15}$ was the least effective (FICI = 0.125–0.5), with a 16-fold reduction in AMB against *C. neoformans* and *C. albicans*, an 8-fold reduction against *M. circinelloides*, and a 4-fold reduction against *A. fumigatus*. Ub, α-lac, and BSA displayed intermediate efficacy, with consistent

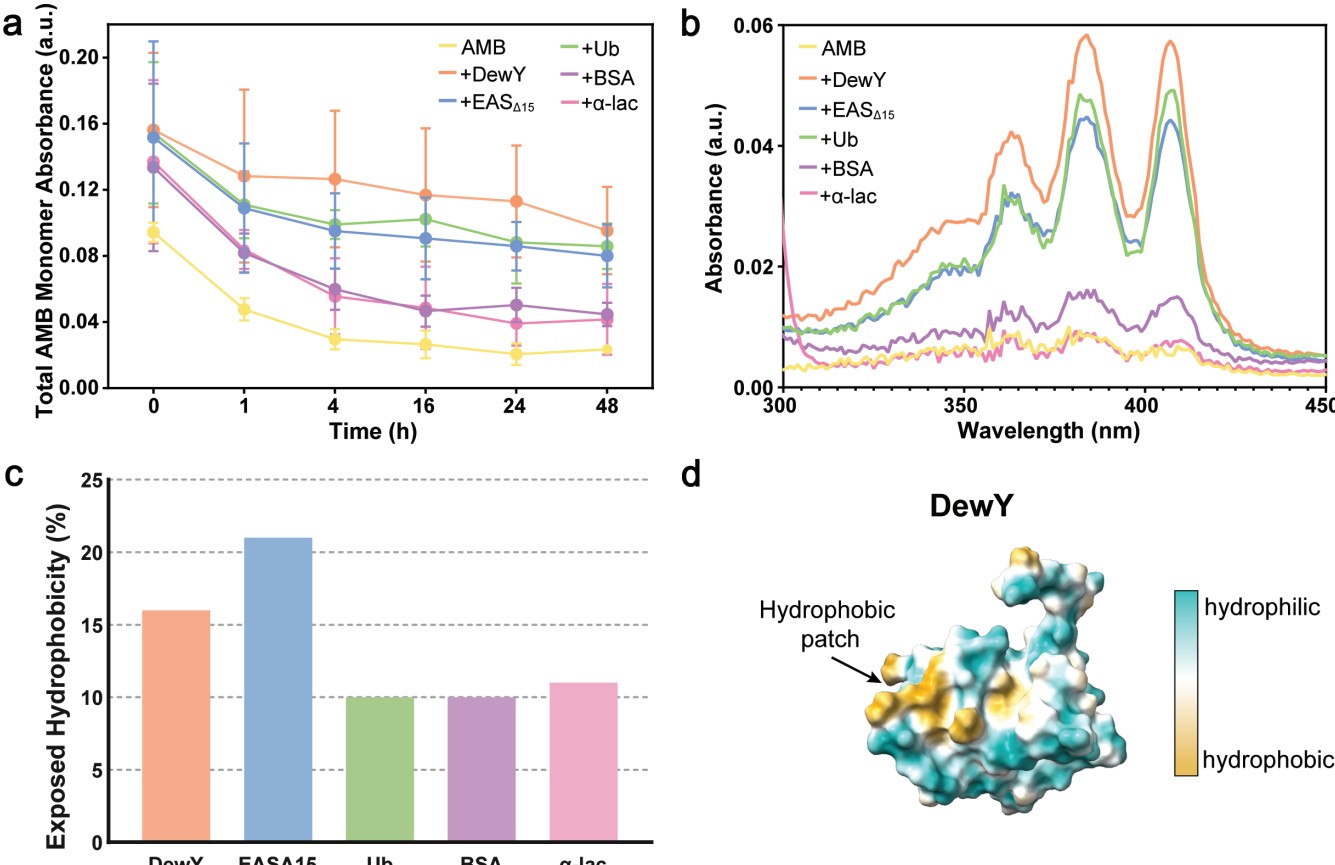

FIG 4   Stabilization of AMB over 48 h in the presence of proteins. (a) Total absorbance of AMB monomers ($A_{409} + A_{385} + A_{365}$) in the absence and presence of the proteins (50 µg/mL) DewY (orange), EAS$_{\Delta 15}$ (blue), Ub (green), BSA (purple) or α-lac (pink) over a 48 h incubation in water. Each point is shown as mean ± standard deviation ($n = 3$). (b) Representative absorbance spectrum of AMB (1 µg/mL) in the absence or presence of the proteins after 48 h. (c) Extent of exposed surface area that is hydrophobic for each protein calculated using Swiss PDB viewer. (d) Surface representation of DewY (PDB ID:2LSH) visualized in ChimeraX, with the surface displayed as molecular hydrophobicity potential with hydrophilic (blue) and hydrophobic (yellow) regions. Example hydrophobic patch indicated with an arrow.

ranking across all species, indicating that the drug-protein interaction primarily drives efficacy rather than species-specific factors.

## AMB-protein induces the same morphological damage to cells and hyphae as AMB alone

Microscopic examination revealed that AMB-protein combinations induced similar morphological changes in fungal cells as AMB alone but at lower drug concentrations (Fig. 5). A treatment concentration of 1/2 MIC or 1/2 FIC was chosen to allow the growth and reproduction of cells while still subjecting them to stress. In *Candida albicans*, AMB and AMB-protein treatments inhibited hyphal production, with AMB + DewY yielding a mix of elongated opaque and round white cells, while AMB + Ub primarily produced the more virulent round white cells (Fig. 5a). In *Cryptococcus neoformans*, AMB-protein treatments reduced the proportion of enlarged cells compared to AMB alone (Fig. 5b). In *Aspergillus fumigatus*, AMB and AMB-protein treatments caused irregularly twisted and highly branched hyphae, with stronger effects seen in AMB + DewY and AMB + Ub (Fig. 5c). Similarly, in *Mucor circinelloides*, AMB-protein treatments reduced sporangia formation, with the remaining sporangia often deformed or swollen (Fig. 5d). Overall, AMB-protein treatments induced the same types of morphological changes as AMB alone but at reduced drug concentrations, indicating that equivalent damage can be achieved using less AMB when it is formulated with proteins.

**TABLE 2** Effect of AMB-protein pairs on fungal pathogens

| Protein | Drug FIC[a] (µg/mL) | AMB fold decrease | Protein FIC[b] (µg/mL) | FICI | Synergy[c] |
|---|---|---|---|---|---|
| *Cryptococcus neoformans* | | | | | |
| AMB alone | 1 | – | – | – | – |
| +DewY | 0.03125 | 32 | 1.5625 | 0.0625 | Synergistic |
| +EAS$_{\Delta15}$ | 0.0625 | 16 | 3.125 | 0.125 | Synergistic |
| +Ubiquitin | 0.03125 | 32 | 1.5625 | 0.0625 | Synergistic |
| +α-Lactalbumin | 0.0625 | 16 | 3.125 | 0.125 | Synergistic |
| +Bovine serum albumin | 0.03125 | 32 | 1.5625 | 0.0625 | Synergistic |
| *Candida albicans* | | | | | |
| AMB alone | 1 | – | – | – | – |
| +DewY | 0.03125 | 32 | 1.5625 | 0.0625 | Synergistic |
| +EAS$_{\Delta15}$ | 0.0625 | 16 | 3.125 | 0.125 | Synergistic |
| +Ubiquitin | 0.0625 | 16 | 3.125 | 0.125 | Synergistic |
| +α-Lactalbumin | 0.0625 | 16 | 3.125 | 0.125 | Synergistic |
| +Bovine serum albumin | 0.0625 | 16 | 3.125 | 0.125 | Synergistic |
| *Aspergillus fumigatus* | | | | | |
| AMB alone | 1 | – | – | – | – |
| +DewY | 0.0625 | 16 | 3.125 | 0.125 | Synergistic |
| +EAS$_{\Delta15}$ | 0.25 | 4 | 12.5 | 0.5 | Synergistic |
| +Ubiquitin | 0.125 | 8 | 6.25 | 0.25 | Synergistic |
| +α-Lactalbumin | 0.125 | 8 | 6.25 | 0.25 | Synergistic |
| +Bovine serum albumin | 0.0625 | 16 | 3.125 | 0.125 | Synergistic |
| *Mucor circinelloides* | | | | | |
| AMB alone | 1 | – | – | – | – |
| +DewY | 0.03125 | 32 | 1.5625 | 0.0625 | Synergistic |
| +EAS$_{\Delta15}$ | 0.125 | 8 | 6.25 | 0.25 | Synergistic |
| +Ubiquitin | 0.03125 | 32 | 1.5625 | 0.0625 | Synergistic |
| +α-Lactalbumin | 0.0625 | 16 | 3.125 | 0.125 | Synergistic |
| +Bovine serum albumin | 0.03125 | 32 | 1.5625 | 0.0625 | Synergistic |

[a]FIC is shown for combinations; MIC is shown for AMB alone.
[b]As proteins had no achievable MIC, the highest concentration tested (50 µg/mL) was used for the calculation of FICI.
[c]≤0.5 = synergistic; >0.5–4.0 = indifferent; >4 = antagonistic.

## DISCUSSION

Since its discovery in the 1950s, AMB has been a cornerstone broad-spectrum antifungal agent. Its primary mechanism involves disrupting and puncturing ergosterol-containing fungal membranes (1), but it can also interact with cholesterol in host cells, which contributes to significant toxicity (30). Additionally, the poor solubility of AMB and tendency to form aggregates present challenges to its pharmacokinetics, where multimolecular aggregates complicate intravenous delivery (29). This study investigated the potential of fungal hydrophobins and other proteins to stabilize AMB in its monomeric state, enhancing its solubility and antifungal efficacy. Hydrophobins, with their relatively large and exposed hydrophobic surfaces, were hypothesized to be uniquely suited to encapsulate or stabilize AMB in aqueous environments (15). However, while hydrophobins were very effective, similar effects were observed with other non-hydrophobin proteins. These findings align with previous studies that reported the potentiation of AMB by individual proteins, such as the milk protein lactoferrin and BSA (9, 35). Our study expands this understanding by demonstrating that protein-mediated stabilization of AMB is not limited to specific proteins but could represent a broader strategy applicable across a range of proteins.

Our results suggest a non-specific stabilization mechanism that involves transient interactions between AMB monomers and exposed hydrophobic patches on proteins, thereby reducing the rate and extent of aggregation, as summarized in Fig. 6. This is supported by the ThT assays, where ThT bound to amyloid rodlets was displaced by AMB.

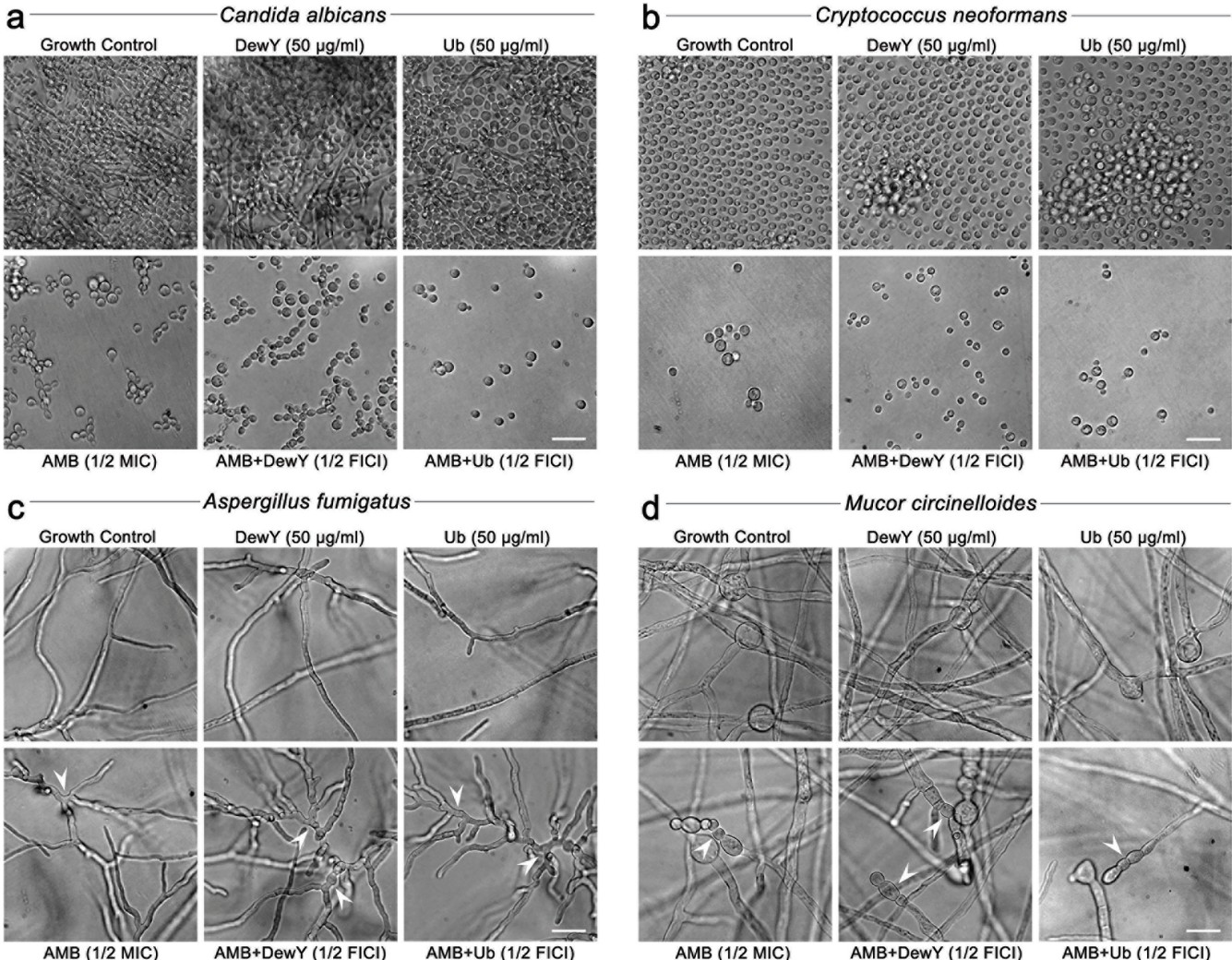

**FIG 5** Morphological changes induced in fungal pathogens by sub-inhibitory concentrations of AMB and AMB-protein pairs. Light microscopy images of (a) *Candida albicans*, (b) *Cryptococcus neoformans*, (c) *Aspergillus fumigatus*, and (d) *Mucor circinelloides* treated with 1/2 MIC of AMB (0.5 µg/mL), 50 µg/mL of DewY or Ub, or 1/2 FIC of AMB + DewY (0.016–0.031 µg/mL AMB) or AMB + Ub (0.016–0.063 µg/mL AMB) after 48 h compared to an untreated growth control. Scale bars = 10 µM. White arrowheads indicate morphological changes following treatment.

The extent of hydrophobicity of the proteins alone does not predict efficacy, as despite its larger hydrophobic surface (Fig. 4d), the AMB-EAS$_{\Delta15}$ combination was not as effective as the AMB-DewY and AMB-BSA pairings. Structural factors like binding groove accessibility may play a role; for example, the BSA surface is known to display several hydrophobic grooves that bind fatty acids, and this may explain its superior performance (36). Of note, protein binding did not alter the underlying mechanism of action of AMB. Microscopy revealed consistent damage by AMB alone and AMB-protein combinations, and there was a consistent ranking in the order of efficacy of AMB-protein pairs against various fungal pathogens. Together, these findings suggest that the individual nature of the protein used to stabilize AMB determines the level of potentiation, which may be influenced by both hydrophobic and structural features.

In the body, AMB binds to various plasma proteins such as lipoproteins and albumin, influencing its distribution, delivery to host tissues, and toxicity (37–39). Consistent with the results presented here, serum albumin has been reported to interact with both AMB and NYS and polyene drugs, increasing the critical aggregation concentration of both and lowering hemolytic activity (39, 40). As well as reducing aggregation, the non-specific binding of AMB to these proteins *in vivo* is thought to reduce the proportion of

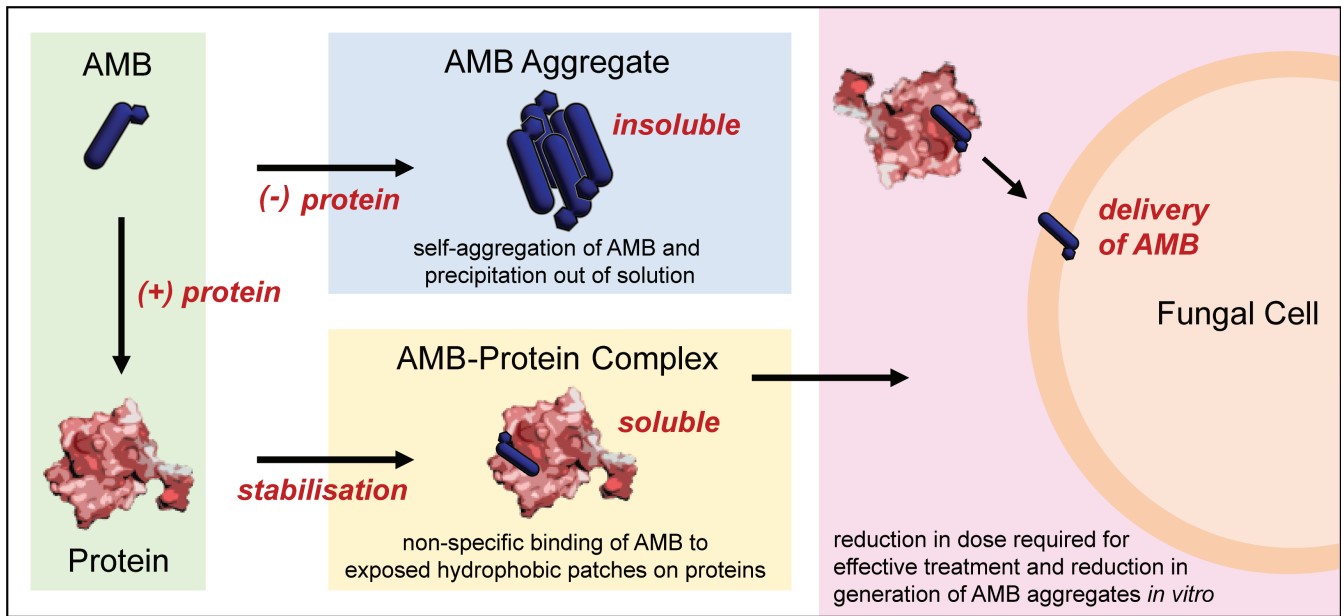

**FIG 6** Schematic illustrating the proposed mechanism of AMB-protein synergy. AMB binds non-specifically to exposed hydrophobic patches on the surface of proteins, stabilizing it in monomeric form in solution.

free unbound AMB present in the body. A study comparing deoxycholate and liposomal formulations of AMB found that liposomal AMB exhibited a greater degree of protein binding, which aligns with a lowered distribution to the kidneys and reduced nephrotoxicity of liposomal AMB and suggests that the free unbound drug drives these processes (41).

Pre-binding of AMB to specific proteins with desirable properties in pharmaceutical formulations could present a promising approach to replicate the beneficial effects of *in vivo* protein interactions, in addition to reducing the required dose by monomer stabilization. Such pre-binding could enhance the effective solubility of AMB in the bloodstream, aid systemic transport, and reduce the levels of free diffusible drug, thereby mitigating off-target effects associated with aggregation. Potential negative effects may include altered pharmacokinetics or drug clearance, possibly impacting efficacy or duration of action (42). For example, reduced AMB mobility might hinder its ability to penetrate certain tissues or cross biological barriers to reach infection sites, or even concentrate the drug at certain sites leading to increased localized toxicity. If these adverse effects could be controlled by selectively pre-binding AMB to proteins with desirable characteristics, toxic effects could be mitigated and targeted delivery could be enhanced.

## Conclusion

This study provides new insights into the role of protein interactions in modulating AMB solubility and stabilizing it in its less toxic monomeric state. As a critical yet toxic antifungal, these findings offer a potential avenue for enhancing the clinical effectiveness of AMB by decreasing its therapeutic dose and reducing the formulation of damaging aggregates. While our study focuses on *in vitro* characterization, further research is needed to assess the safety and efficacy of particular AMB-protein complexes in host cells and animal models. Investigating their stability and functionality under physiological conditions, their pharmacokinetics, and their ability to reach infection sites effectively will be essential for translation into clinical applications. Exploring protein-engineering approaches to optimize the AMB-protein interactions could further improve drug delivery and reduce off-target effects. These findings may also inform

broader applications for protein-drug interactions in other therapeutic areas where drug solubility and toxicity are major limitations.

## ACKNOWLEDGMENTS

This study was funded by The University of Sydney Centre for Drug Discovery Innovation and the Sydney Infectious Diseases Institute. C.L.J. was supported by the Australian Research Council Discovery Project grant (DP200102463) awarded to M.S. The funders played no role in study design, data collection, analysis and interpretation of data, or the writing of this article. The authors acknowledge the facilities and the scientific and technical assistance of staff within Sydney Microscopy & Microanalysis Core Research Facility at the University of Sydney. We acknowledge and pay respect to the Gadigal people of the Eora Nation, the traditional owners of the land on which we researched and collaborated at the University of Sydney.

K.E.F. and C.L.J. conceived and designed the experiments, collated and analyzed the data, and wrote the manuscript. K.E.F. produced antifungal susceptibility testing, drug interaction, and fungal morphology data. C.L.J. and B.C.W. produced all protein + drug samples tested in this study, in addition to the solubility (UV-visible) and AMB binding (thioflavin T and transmission electron microscopy) data. D.A.C. and M.S. assisted with data analysis and critically reviewed and edited the manuscript. All authors read and approved the final manuscript.

## AUTHOR AFFILIATIONS

[1]School of Life and Environmental Sciences, University of Sydney, Sydney, New South Wales, Australia

[2]Sydney Institute for Infectious Diseases, University of Sydney, Sydney, New South Wales, Australia

[3]School of Medical Sciences and Sydney Nano, University of Sydney, Sydney, New South Wales, Australia

## AUTHOR ORCIDs

Kenya E. Fernandes http://orcid.org/0000-0002-2912-4360
Caitlin L. Johnston http://orcid.org/0009-0006-1062-9680
Brayden C. Williams http://orcid.org/0009-0007-2713-8084
Dee A. Carter http://orcid.org/0000-0002-4638-0059
Margaret Sunde http://orcid.org/0000-0002-0150-3203

## FUNDING

| Funder | Grant(s) | Author(s) |
| --- | --- | --- |
| The University of Sydney Centre for Drug Discovery Innovation and the Sydney Infectious Diseases Institute | | Kenya E. Fernandes |
| | | Dee A. Carter |
| | | Margaret Sunde |
| Department of Education and Training \| Australian Research Council (ARC) | DP200102463 | Margaret Sunde |

## AUTHOR CONTRIBUTIONS

Kenya E. Fernandes, Conceptualization, Data curation, Formal analysis, Funding acquisition, Investigation, Methodology, Project administration, Resources, Supervision, Validation, Visualization, Writing – original draft, Writing – review and editing | Caitlin L. Johnston, Conceptualization, Data curation, Formal analysis, Investigation, Methodology, Resources, Supervision, Validation, Visualization, Writing – original draft, Writing – review and editing | Brayden C. Williams, Formal analysis, Investigation, Methodology,

Validation, Writing – review and editing | Dee A. Carter, Conceptualization, Funding acquisition, Resources, Supervision, Writing – review and editing | Margaret Sunde, Conceptualization, Formal analysis, Funding acquisition, Methodology, Project administration, Resources, Supervision, Writing – review and editing

## DATA AVAILABILITY

All data generated or analyzed during this study are included in this published article.

## ADDITIONAL FILES

The following material is available online.

Open Peer Review

**PEER REVIEW HISTORY (review-history.pdf).** An accounting of the reviewer comments and feedback.

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
