## [Reviewer comments · Microbiology Spectrum]

Microbiology Spectrum

Protein-mediated stabilisation of Amphotericin B increases its efficacy against diverse fungal pathogens

Kenya Fernandes, Caitlin Johnston, Brayden Williams, Dee Carter, and margaret sunde

Corresponding Author(s): Kenya Fernandes, The University of Sydney

Review Timeline:

Submission Date:	March 14, 2025
Editorial Decision:	March 21, 2025
Revision Received:	March 23, 2025
Accepted:	March 24, 2025

Editor: Ping Ren

Reviewer(s): The reviewers have opted to remain anonymous.

Transaction Report:

DOI: <https://doi.org/10.1128/spectrum.00686-25>

Re: Spectrum00686-25 (**Protein-mediated stabilisation of Amphotericin B increases its efficacy against diverse fungal pathogens**)

Dear Dr. Kenya E Fernandes:

Thank you for the privilege of reviewing your work. Below you will find my comments, instructions from the Spectrum editorial office, and the reviewer comments.

The authors have responded to the reviewers' comments and improved on the manuscript. I am pleased to inform you that your manuscript has been editorially accepted for publication. However, there are a few additional questions in the submission form that need to be answered before the final decision. Once these are completed, please return your submission so that I can move your paper forward to acceptance.

Sincerely,
Ping Ren
Editor
Microbiology Spectrum

Re: Spectrum00686-25R1 (**Protein-mediated stabilisation of Amphotericin B increases its efficacy against diverse fungal pathogens**)

Dear Dr. Kenya E Fernandes:

Your manuscript has been accepted, and I am forwarding it to the ASM production staff for publication. Your paper will first be checked to make sure all elements meet the technical requirements. ASM staff will contact you if anything needs to be revised before copyediting and production can begin. Otherwise, you will be notified when your proofs are ready to be viewed.

Sincerely,
Ping Ren
Editor
Microbiology Spectrum